# The Low-FODMAP Diet, IBS, and BCFAs: Exploring the Positive, Negative, and Less Desirable Aspects—A Literature Review

**DOI:** 10.3390/microorganisms11102387

**Published:** 2023-09-25

**Authors:** Maroulla D. Nikolaki, Arezina N. Kasti, Konstantinos Katsas, Konstantinos Petsis, Sophia Lambrinou, Vasiliki Patsalidou, Sophia Stamatopoulou, Katerina Karlatira, John Kapolos, Konstantinos Papadimitriou, Konstantinos Triantafyllou

**Affiliations:** 1Department of Nutrition and Dietetics, ATTIKON University General Hospital, 12462 Athens, Greece; maroullanikolaki@gmail.com (M.D.N.); kastiare@med.uoa.gr (A.N.K.); katkonstantinos@gmail.com (K.K.); kostas.petsis@hotmail.com (K.P.); vasilikipatsalidou@gmail.com (V.P.); sophia.stamatopoulou@hotmail.com (S.S.); k.karlatira@gmail.com (K.K.); 2Department of Nutrition and Dietetics Sciences, Hellenic Mediterranean University, 72300 Crete, Greece; 3Institute of Preventive Medicine Environmental and Occupational Health Prolepsis, 15125 Athens, Greece; 4Department of Clinical Nutrition & Dietetics, General Hospital of Karpathos “Aghios Ioannis o Karpathios”, 85700 Karpathos, Greece; sophialambrinou@gmail.com; 5Department of Food Science and Technology, University of Peloponnese, 24100 Kalamata, Greece; i.kapolos@uop.gr; 6Laboratory of Food Quality Control and Hygiene, Department of Food Science and Human Nutrition, Agricultural University of Athens, 11855 Athens, Greece; kpapadimitriou@aua.gr; 7Hepatogastroenterology Unit, 2nd Department of Internal Propaedeutic Medicine, Medical School, National and Kapodistrian University of Athens, ATTIKON University General Hospital, 12462 Athens, Greece

**Keywords:** branched-chain fatty acids, short-chain fatty acids, irritable bowel syndrome, low-FODMAP diet, gut microbiota, inflammation

## Abstract

The literature about the association of branched short-chain fatty acids (BCFAs) and irritable bowel syndrome (IBS) is limited. BCFAs, the bacterial products of the catabolism of branched-chain amino acids, are proposed as markers for colonic protein fermentation. IBS is a gastrointestinal disorder characterized by low-grade inflammation and intestinal dysbiosis. The low-FODMAP diet (LFD) has increasingly been applied as first-line therapy for managing IBS symptoms, although it decreases the production of short-chain fatty acids (SCFA), well known for their anti-inflammatory action. In parallel, high protein consumption increases BCFAs. Protein fermentation alters the colonic microbiome through nitrogenous metabolites production, known for their detrimental effects on the intestinal barrier promoting inflammation. **Purpose**: This review aims to explore the role of BCFAs on gut inflammation in patients with IBS and the impact of LFD in BCFAs production. **Methods**: A literature search was carried out using a combination of terms in scientific databases. **Results**: The included studies have contradictory findings about how BCFAs affect the intestinal health of IBS patients. **Conclusions**: Although evidence suggests that BCFAs may play a protective role in gut inflammation, other metabolites of protein fermentation are associated with gut inflammation. Further research is needed in order to clarify how diet protein composition and, consequently, the BCFAs are implicated in IBS pathogenesis or in symptoms management with LFD+.

## 1. Introduction

Irritable bowel syndrome (IBS) is a highly prevalent disorder characterized by abdominal pain and changes in bowel habits. Its etiology is still unknown, although patients often report an infectious or traumatic event triggering the onset of symptoms. The pathogenetic mechanisms include, among others, increased intestinal permeability, intestinal dysbiosis, visceral hypersensitivity, gut–brain axis dysregulation, low-grade intestinal inflammation, and psychological stress [1]. Additionally, alterations in intestinal fermentation may lead to abnormalities, such as intraluminal excessive gas production and altered motility, which are common in IBS. Intraluminal intestinal fermentation by colonic bacteria produces gases such as hydrogen and carbon dioxide, short-chain fatty acids (SCFAs), and branched-chain fatty acids (BCFAs) as secondary byproducts. SCFAs and BCFAs are natural acids and cause a significant pH drop in the intestinal lumen (the intraluminal pH varies depending on the intestinal segment and ranging from 5.5–7.5 in the cecum/right colon, and to 6.1–7.5 in the left colon and rectum) [2,3].

Fatty acids are the most important components of bacterial membrane lipids. BCFAs are a group of saturated lipids presenting in many organisms affecting multiple signaling pathways. BCFAs were first recognized as significant nutrients for the gastrointestinal tract (GI) health in the context of vernix caseosa, the white waxy substance that develops during the last trimester of human fetuses [4]. BCFAs constitute a considerable component of many gut bacteria (~15% of phyla) [5]. Although they take part in numerous biochemical procedures, BCFAs are not satisfactorily explored, and research in humans is scarce [6]. BCFAs, mainly isobutyrate, isovalerate, and 2-methyl butanoate, contribute to as much as 5% of total SCFA production and derive from the metabolism of valine, leucine, and isoleucine, respectively (all belong to essential amino acids) [7]. BCFAs are highly abundant in the cecum and colon and their levels in fecal samples have inversely correlated to fiber (especially insoluble) consumption [7,8]. While the responses to SCFAs in the gut are mediated through the free fatty acid receptors 2 (FFA2) and 3 (FFA3) binding to G-protein (GPR41 and GPR43), little is known about the metabolism of BCFAs in host physiology. Studies have highlighted their role in the ionic exchange. More specifically, isobutyrate can increase the Na^+^/H^+^ exchanger, suggesting that this fatty acid regulates the absorption of Na in the colon [9]. Furthermore, the abundance of BCFAs has been linked to decreased levels of *Firmicutes* and higher levels of *Coprococcus*, *Blautia*, and *Bacteroidetes* during high-protein diets [10,11].

Interconnections between microbes, foods, and hosts in the human gut form a complex ecological network. Gut microbiota produces metabolites through the fermentation of nondigested food components, and specific metabolites are produced depending on the type of food the individual consumes [12]. Food is one of the factors that trigger symptoms in IBS patients, and specific dietary patterns are the first-line therapeutic approach. The low-FODMAP diet (LFD) is gaining ground as the most well-documented dietary intervention in reducing these symptoms [13].

FODMAP stands for fermentable oligo-, di-, mono-saccharides, and polyols, a large class of small nondigestible carbohydrates containing up to 10 sugars. Those are poorly absorbed in the small bowel and they are potential triggers for exacerbating abdominal symptoms in IBS patients [14]. The intolerance to FODMAPs causes luminal distension while other metabolic products, such as pathogen-associated molecules, may induce pain symptoms, particularly in IBS patients with visceral hypersensitivity [15]. The core mechanism of FODMAPs is related to their ability to attract water, thus increasing the amount of fluids in the intestines and contributing to their osmotic activity [16]. Furthermore, they constitute food for the gut microflora, which ferment them and increase gas production. It has been well documented that FODMAPs activate Meissner’s plexus, modulating the neuroenteric sensory transmission stimulating intestinal secretion and motility, and accelerating transit time [17]. Additionally, the effects of FODMAPs on gut microbiota, intestinal barrier, immune response, and visceral sensation may also contribute to symptom generation [18]. LFD reduces proinflammatory interleukins (ILs) IL-6 and IL-8 serum levels, *Actinobacteria*, *Bifidobacterium*, and *Faecalibacterium prausnitzii* abundance, and SCFAs and n-butyric acid in the feces [19,20,21,22], while individuals with IBS that are responders of LFD have higher proportions of *Bacteroidaceae*, *Erysipilotrichaceae*, and *Clostridiales* species, with a greater capacity for saccharolytic metabolism [23].

The flow of undigested carbohydrates, especially fibers, into the gut is associated with beneficial effects, as they are preferentially used by many bacterial species as the main energy source to produce SCFAs. SCFAs, especially butyrate, have been related to improved epithelial barrier function and a decrease in pH [24,25]. Restricting fermentable substrates for saccharolytic gut bacteria reduces SCFA production [13]. When saccharolytic fermentation (carbohydrate) is reduced and protein fermentation significantly occurs, elevated concentrations of BCFAs are detected across the colon [26]. The proteolytic activity in the large intestine has been mainly attributed to the genera *Bacteroides*, *Propionibacterium*, *Clostridium*, *Streptococcus*, *Fusobacterium*, and *Lactobacillus*. *Bacteroides* spp. secreting proteases act like elastase and, in the case of their abundance (or overgrowth), they may degrade maltase and sucrase enzymes in the enterocyte brush borders [27].

Intestinal inflammation status is closely related to gut health, and low-grade inflammation has been detected in the large bowel of patients suffering from IBS. Moreover, consumption of specific food may lead to alteration of the abundance of various microorganisms. The way that food is processed could also lead to inflammation. For example, advanced glycosylation end products in milk, created after heating, affect the composition of the gut microbiome, which impacts intestinal function [28]. Protein fermentation results in the formation, among others, of ammonia, phenolic and indolic compounds, hydrogen sulfide, nitric oxide (NO), and biogenic amines, which have been associated with detrimental effects on gut health and how they have been implicated in the pathogenesis of colorectal cancer, inflammatory bowel disease (IBD), and IBS through the induction of a chronic inflammatory status in the intestine [25].

BCFAs have not attracted attention like SCFAs, even though they may have a crucial role in the gut milieu and could be considered potential markers of microbial metabolism [29]. Butyrate is the main energy source of colonocytes, providing 70–80% of their energy requirements and regulating colonic homeostasis, but BCFAs have the potential to be oxidized when butyrate is not available, and isobutyrate could be used as an alternative energy source [2,30,31]. Moreover, like butyrate, BCFAs are potent histone deacetylase (HDAC) inhibitors, and, therefore, their function in regulating host cells could be similar to those of butyrate [32]. Isovalerate has also been reported as a potent enterochromaffin (EC) cell stimulus. EC cell stimulation leads to voltage-gated calcium (Ca^2+^) channel-dependent serotonin (5-HT) release and forms synaptic-like contacts with 5-HT-expressing nerve fibers. Therefore, isovalerate modulates 5-HT-expressing primary afferent nerve fibers through the endothelial cells [8]. These findings confirm the way microbial-derived metabolites directly correlate with the enteric nervous system.

### The Role of BCFAs in Intestinal Inflammation

Intestinal inflammation disturbs normal growth in humans and animals and leads to bowel diseases [33]. Many commensal bacteria utilize BCFAs to survive in the varying milieus. Modulating membrane fluidity is essential for bacterial survival in a variety of environments, and many microorganisms use BCFAs in their membranes to modulate biophysical procedures [34]. BCFAs are taken up and incorporated into enterocyte membranes where they modulate the inflammatory response [35,36,37]. In detail, as shown in Figure 1, BCFAs affect the cell membrane’s fluidity: with high BCFAs concentrations, membrane falls into a disordered state, allowing better transport, membrane protein structure and functionality, and cellular signal transduction and trafficking. Expectedly, a higher concentration of SCFAs in the membrane is noticed [38]. During this transition, the membrane has larger hydrophobic thickness and lower mean lateral area occupied by lipids. The membrane enters the usual state as fatty acid acyl chains are impenetrable together. Specifically, the membrane coagulates and loses fluidity, impeding regular cellular processes such as active and passive transport, membrane protein structure, and signal transport. Thus, BCFAs’ capacity to produce large amounts of SCFAs may modulate and decrease intestinal inflammation [38].

While the three SCFAs (acetate, butyrate, and propionate) produced by fermented nondigestible carbohydrates might inhibit inflammation and exert anti-inflammatory effects via the NF-κB/NLRP3 signaling pathway [39], the anti-inflammatory potential of dietary BCFAs has been demonstrated in vitro in the human-derived intestinal cell line Caco-2. Exposure of Caco-2 cells to BCFAs decreased the lipopolysaccharide-induced gene expression of proinflammatory mediators (i.e., IL-8, TLR-4, and NF-κB), while shorter-chain BCFAs (branched short-chain carboxylic acids are iso-14:0 and iso-16:0 -series, derived from valine, iso-15:0 and iso-17:0-series from leucine, and anteiso-15:0 and anteiso-17:0-series from isoleucine) incorporated into phospholipids similarly to rat infants’ ileum, and improved the lipopolysaccharide-induced reduction in cell viability [37,40].

BCFAs increase protein SUMOylation (small ubiquitin-like modifier, a crucial ubiquitin-like modification involved in numerous intestinal functions) in intestinal cell lines in a pH-dependent manner. Ezzine et al. assessed the role of BCFAs in the inflammatory response and intestinal epithelial integrity in Caco-2 cell cultures. The results revealed that fatty acids produced by gut bacteria could regulate intestinal physiology by modulating SUMOylation and diminishing host inflammatory responses triggered by the gut microbiota. In addition to SUMOylation, BCFAs could inhibit the NF-κB pathway, decreasing proinflammatory cytokine expression and promoting intestinal epithelial integrity [2].

Necrotizing enterocolitis (NEC) is an inflammatory disease in the GI of premature infants, which is a major cause of morbidity, with an estimated rate of death of 20–30% [5]. Ran-Ressler et al. evaluated the effects of BCFAs on NEC in neonatal Sprague Dawley rats. BCFA-fed rats had a lower incidence of NEC and a higher expression of anti-inflammatory cytokine IL-10. BCFAs influenced the gut microbiota composition of the rat’s intestine, promoting *Bacillus subtilis* abundance, which was associated with lower levels of proinflammatory and higher levels of anti-inflammatory cytokines, as well as better immune system performance and overall animals’ condition (based on weight gain and food intake) [41].

According to Russel [42] and her colleagues, high-protein and low-carbohydrate diets increase the BCFA levels, the phenylacetic acid concentrations, and N-nitroso compounds. That dietary pattern correlated with reduced abundance of *Roseburia*/*Eubacterium rectale*, a beneficial bacterium for gut health. Furthermore, high levels of *Faecalibacterium prausnitzii,* which is well known for its anti-inflammatory effects on the intestinal mucosa, were detected [42]. Following a high-protein formula consumption, female piglets displayed decreased levels of *F. prausnitzii* and BCFAs [43]. Moreover, the results highlighted that BCFAs have a dose-dependent protective effect against the proinflammatory cytokines tumor necrosis factor-α (TNFα) and interferon (IFNγ). High doses of BCFAs reduce transepithelial electrical resistance (TEER), which is the measurement of electrical resistance across a cellular monolayer. Decreased levels of TEER are related to loosening of tight junctions. BCFAs appear to exert their anti-inflammatory effect through the upregulation of the Zonulin (ZO-1) network and claudin-1 [43]. Alterations of tight junction proteins were associated with visceral hypersensitivity, abdominal pain, and mast cell activation. The increased GI permeability results in bacterial translocation through the mucosal barrier, influencing immune responses and contributing to the low-grade inflammation in IBS [44].

In a piglet model, fed with Lentinan (a type of mushroom polysaccharide), challenged with *Escherichia coli,* lipopolysaccharide-induced, isobutyrate production in cecal digesta was positively related to *Faecalibacterium*, *Prevotella*, and unclassified *Ruminococcaceae*, while the high production of isobutyrate and isovalerate was associated with decreased proinflammatory cytokines (TNF-a, interleukin-1β, and interleukin-6) [45].

In vitro studies using protein-fermenting bacteria detected BCFA production when the microorganisms were grown with peptides at pH of 6.8. On the contrary, the presence of starch at pH of 5.5 decreased the formation of BCFA in these cultures [29]. In pigs fed with low dietary fiber, He et al. evaluated the effect of less rapidly fermented fibers like resistant starch on protein fermentation by inocula from the large intestine in in vitro cultivation. The researchers observed that the resistant starch weakens protein fermentation by influencing gut microbiota and lowering BCFA levels in the cecum and colon [46].

## 2. Literature Search

In this review, we aimed to explore the role of BCFAs on gut inflammation in patients with IBS and the impact of LFD in BCFAs production. We assessed preclinical and human studies during 2010–2023. We performed a literature search in the PubMed and Google Scholar databases for articles written in the English language. We used evidence from original articles, excluding reviews, abstracts, conference presentations, editorials, and study protocols and included studies identified by the manual search of the reference lists of the aforementioned articles (Figure 2). We used a combination of MESH and keyword search terms “IBS”, “irritable bowel syndrome”, “low FODMAP diet”, “isovalerate”, “isobutyric” and “Branched-chain fatty acids” (accessed on 7 July 2023). All included studies (preclinical, clinical) should have investigated how BCFAs levels change in individuals with IBS. Furthermore, we included articles that reported the correlation between BCFAs LFD as the first-line treatment for patients with IBS. IBD studies were excluded. Two reviewers (M.N., A.K.) independently extracted the data from eligible studies. Any disagreement was solved by consensus.

### 2.1. BCFAs & IBS

#### Human Studies, Table 1

In a metabolomic study, nuclear magnetic resonance (NMR) spectroscopy was used to identify complex mixture metabolites present in fecal samples to explore differences among IBS (*n* = 10), UC (*n* = 13), and controls (*n* = 22). The results showed lower BCFA production in the IBS group compared to controls [47]. Some years later, Farup and his colleagues assessed 25 patients with IBS and 25 healthy controls in a case–control study to evaluate the properties of fecal metabolites as diagnostic biomarkers for IBS. Stool samples were analyzed with gas chromatography for SCFAs and BCFAs (isobutyric and isovaleric acid). Their results indicated that fecal SCFA levels were highly significant and could be valid and reliable biomarkers for IBS diagnosis, but no significant differences were observed in BCFAs between the two groups; isovaleric acid on its own did not appear to be a robust diagnostic tool [48].

**Table 1 microorganisms-11-02387-t001:** Human studies in IBS patients and BCFA levels.

References	Type of Study	Subjects	Samples	Outcomes
Le Gall et al., 2011 [47]	Cohort	UC patients (*n* = 13). IBS patients (*n* = 10). Healthy controls (*n* = 22).	Feces	↑ 2-methylbutyrate, isobutyrate, isovalerate in control vs. IBS group.
Farup et al., 2016 [48]	Case–control	IBS patients (All subtypes, *n* = 25). Healthy controls (*n* = 25).	Feces	No statistically significant difference in isovaleric or isobutyric between IBS and control.
Zhang et al., 2019 [49]	Case–control	IBS-D patients (*n* = 30). Healthy controls (*n* = 15).	Feces	↑ Isobutyrate IBS-D (no statistically significant different). ↑ Isovalerate in IBS-D correlated with severity of abdominal pain.

UC: ulcerative colitis; IBS-D: irritable bowel syndrome with diarrhea.

In another case–control study, the fecal metabolites composition and the role of metabolites were investigated in 30 IBS patients with diarrhea (IBS-D) and 15 healthy controls. Data showed that isobutyric acid levels were higher in the stools of patients with IBS, whereas isovalerate levels positively correlated with the severity or frequency of abdominal pain. Both isovaleric and isobutyric acids were associated with visceral hypersensitivity and contributed to abdominal pain. Furthermore, isohexanoate was significantly related to the severity or frequency of abdominal pain in IBS-D patients [49].

Another team studied the changes in fecal fatty acids after fecal microbiota transplantation (FMT) in IBS (all subtypes) patients. One hundred and forty-two IBS patients were divided into three groups: placebo (own feces), 30 g (superdonor feces), and 60 g (superdonor feces). In responders of the 60 g FMT group, fecal levels of isovaleric and isobutyric rose overall. Isobutyric levels increased in IBS-D and IBS with constipation (IBS-C), but not in IBS mixed (IBS-M) patients. On the contrary, isovaleric acid levels increased only in the IBS-D patients, while there was no statistical significance between BCFAs and IBS-SSS scores. Authors tried to explain the differences in BCFAs levels between IBS subtypes with the following speculation: “due to the location of the fermentation of BCFAs in the proximal colon, and the rapid absorption of BCFAs by epithelial cells, the intestinal transit time could be the causative factor for their low levels” [50].

Moreover, higher concentrations of BCFAs were positively associated with longer colonic transit time (CTT), and longer CTT was also associated with increased proteolytic fermentation in healthy adults, while higher levels of SCFA were related to shorter CTT [11]. A possible hypothesis is that LFD and higher levels of BCFA could be indicated for IBS-D management, while a high-FODMAP diet and higher SCFA levels could be suitable for IBS-C. Certainly, CTT is affected by a plethora of factors, such as sex, age, stress, body mass index, colonic anatomy, treatment, and gut hormones, among others [11]. Nonetheless, diet remains one of the most important keys for the shape of the host’s gut microbiota, GI motility, CTT, and fecal bulk.

### 2.2. BCFAs, Inflammation, and Low-FODMAP Diet

#### 2.2.1. Preclinical Trials (Table 2)

In 2019, Tuck et al. [51] performed experiments in mice with dextran sodium sulfate (DSS)-induced colitis during the inflammatory phase and after treatment. Animals were divided into three groups: two control treatments (“negative-control” and “positive-control”; with and without inflammation, respectively) and a “post-inflammatory” treatment group that mimicked quiescent IBD with IBS-like symptoms. After the recovery, mice were randomized to 2-week low-(0.51 g) or high-FODMAP (4.10 g) diets, respectively. In the positive-control and post-inflammatory treatment groups following the LFD, total levels of stool BCFAs were higher compared with those of the negative controls; statistical significance was reached for isobutyric and isovaleric acid. The results suggested that the higher proteolytic fermentation occurred in the LFD group. Considering the alterations of inflammatory markers, Myeloperoxidase (MPO) activity was lower in the negative-control group, in contrast to its higher levels in the positive-control group, regardless of the FODMAPs content. The research team created a histological score to evaluate the inflammatory status in the colon. The negative controls scored 0, but positive controls had higher scores only in the high-FODMAP diet group. Even if compared with the post-inflammatory group, positive control animals fed with a high-FODMAP diet had significantly higher histological scores. However, no significant differences were observed between low- and high-FODMAP diets. There were no significant changes regarding TNF-α, Granulocyte-macrophage colony-stimulating factor (GM-CSF), IL-10, and IL-4. However, there was a trend for higher TNF-α levels in the positive control versus negative control groups, respectively. Additionally, IL-1β was significantly increased in positive controls and post-inflammatory groups and in mice that consumed a high-FODMAP diet. As far as we know, this is the first preclinical study with an inflammatory-induced condition in the colon and a dietary intervention with LFD, which aimed to measure the BCFA levels and simultaneously some inflammatory biomarkers [51]. A significant increase in BCFAs, specifically isovalerate and isobutyric, was noticed in the LFD group, indicating a strong correlation between LFD and proteolytic fermentation. On the other hand, reducing dietary FODMAPs neither exacerbated nor improved the inflammatory biomarkers [51].

**Table 2 microorganisms-11-02387-t002:** Preclinical trials with FODMAP diet and BCFA levels.

References	Subjects	Intervention	Samples	Outcomes
Tuck et al., 2019 [51]	Mice (*n* = 35)	**3 groups** Positive control + low/high FODMAP (*n* = 12). Negative control + low/high FODMAP (*n* = 12). Post-inflammatory + low/high FODMAP (*n* = 11).	Feces	In positive control and post-inflammatory groups that followed LFD: ↑ BCFAs; ↑ Isovalerate and isobutyric.
Tuck et al., 2020 [52]	Mice (*n* = 40)	**4 groups** Group A (*n* = 10) *LabDiet 5066—Sacrificed at baseline.* Group B (*n* = 10) *LabDiet 5066.* Group C (*n* = 10) *ResearchDiets AIN93G (lower FODMAP content)*. Group D (*n* = 10) *LabDiet 5001 (higher FODMAP content).*	Feces	Group C: ↑ BCFAs; ↑ Isovalerate and isobutyric.

The same research team evaluated commercially available rodent diets across research institutions. Forty mice were randomized into four groups to assess the dietary impact on gut microbiota, SCFAs, and BCFAs profiles. Animals in Group A were euthanized at baseline (controls), mice in Group B received the breeding institution chow, mice in Group C received a low-gluten and LFD, and Group D consumed a high-gluten and FODMAP diet. In the LFD group, the BCFA stool levels were higher compared with those in the high-FODMAP group. More specifically, isovalerate and isobutyric acids levels were highest in the LFD group. The explanation is based on protein fermentation and reflects a trend for increased protein metabolism in low-carbohydrate diets. Furthermore, luminal pH was higher in group C, indicating lower rates of carbohydrate fermentation [52]. This study was conducted in healthy animals with the same dietary model (intervention with formulas high and low in FODMAP) and their findings agreed with the previous study [51], confirming that LFD is associated with increased levels of BCFA and proteolytic fermentation.

#### 2.2.2. Clinical Trials (Table 3)

Three clinical trials [53,54,55] assessed the impact of LFD on IBS patients and measured BCFA production, while the last one provided a high-FODMAP diet [56] on IBS patients and measured the same fecal metabolites. Unfortunately, none of them measured inflammatory biomarkers.

**Table 3 microorganisms-11-02387-t003:** Clinical trials with FODMAP diet and BCFA levels.

References	Subjects	Intervention	Samples	Outcomes
Halmos et al., 2014 [53]	IBS patients (all subtypes, *n* = 27), healthy controls (*n* = 6)	Habitual diet	Feces	↑ Isobutyrate and isovalerate in healthy controls.
		**2 groups** LFD (3.05 g FODMAP). Australian diet (23.7 g).	Feces	No statistically significant difference in BCFAs among the groups.
Wilson et al., 2020 [54]	IBS patients (*n* = 69)	**3 groups** Sham diet with placebo supplement (control) (*n* = 23). LFD supplemented with placebo (*n* = 22). LFD supplemented with 1.4 g/d B-GOS (*n* = 24).	Feces	No statistically significant difference in isobutyrate and isovalerate in LFD with or without supplements of B-GOS.
Zhang et al., 2021 [55]	IBS-D patients (*n* = 100)	**2 groups** LFD (*n* = 51). TDA (*n* = 49).	Feces	In LFD group: ↑ Isobutyrate and isovalerate.
Nordin et al., 2023 [56]	IBS patients (*n* = 103)	**3 groups** 1. Placebo–gluten–FODMAPs (*n* = 35). 2. FODMAPs–placebo–gluten (*n* = 33). 3. Gluten–FODMAPs–placebo (*n* = 35).	Feces and plasma	No statistically significant difference in isobutyrate among groups (feces). ↓ Isovalerate after gluten vs. placebo. ↓ Isobutyrate after FODMAPs compared to the placebo (plasma).

LFD: low-FODMAP diet; TDA: traditional dietary advice; IBS-D: irritable bowel syndrome with diarrhea; BCFAs: branched-chain fatty acids.

Halmos et al. [53] assessed the effects of an LFD versus a typical Australian diet on fecal biomarkers. This study included 33 participants (27 with IBS and 6 healthy controls). Volunteers followed two diets differing in FODMAP content (LFD contained 3.05 g, whereas the typical Australian diet contained 23.7 g of FODMAPs, respectively). At baseline, isovaleric and isobutyric levels were lower in IBS patients compared to controls, though no difference was noticed during the dietary intervention [53].

In 2020, 69 patients who fulfilled IBS Rome III criteria were randomized into three groups: a sham diet with a placebo supplement (control), or LFD supplemented with either placebo (LFD), or 1.4 g/d β-Galactooligosaccharide (LFD/B-GOS) for four weeks. Fecal BCFAs were measured at baseline and the end of intervention. Isobutyric and isovalerate acid levels did not significantly increase at the end of the LFD, with or without supplements of B-GOS [54].

Therapeutic effects of LFD in comparison with traditional dietary advice (TDA), as well as the factors associated with favorable outcomes, were examined in Chinese patients with IBS-D (Rome III criteria). One hundred and eight patients were randomized to LFD or traditional TDA. BCFAs were measured at baseline and three weeks after initiation of the intervention. Patients in the LFD, but not in the TDA, group showed improvement in symptoms within the first week. Furthermore, BCFAs, specifically isovalerate and isobutyric, increased in the LFD group [55].

Recently, Nordin and her team investigated the effects of different dietary patterns on fecal microbiota, fecal fatty acids, and plasma metabolome in IBS symptoms. One hundred and three IBS patients were randomized into three groups in this double-blinded, placebo-controlled trial. Each group followed all the dietary plans (placebo, gluten, and FODMAP) but in a different sequence. Placebo consisted of 18 g of sucrose, gluten intervention contained 17.3 g of gluten, and the daily dose of FODMAP intake was 50 g. During the intervention, patients filled in questionnaires, and they underwent blood and fecal analyses and anthropometric measurements periodically. Results showed a reduction in plasma levels of isobutyrate in the FODMAP group compared to the placebo, while in feces, a decrease was observed in isovalerate after the gluten diet. Furthermore, researchers highlighted that isovalerate levels in plasma are linked to isovalerate and isobutyrate levels in feces. Regarding fecal microbiota, an unknown *Christensenellaeae* and *Coprostanotigenes* genus was negatively related to isovalerate and isobutyrate [56].

The results of the last two studies [55,56] are in contrast with the previous research results [53,54] that enrolled all subtypes of IBS patients. Thus, a logical hypothesis for this differentiation is the inclusion of different IBS subtypes in the studies [11].

To summarize, evidence from preclinical and clinical studies shown in Table 2 and Table 3 is conflicting: Studies conducted in experimental animal models of IBS detected raised levels of BCFAs following LFD. Human studies in IBS patients, with different subtypes, reported different levels of fecal of BCFAs. Moreover, there was no consistent correlation of IBS symptoms, like abdominal pain, with the measured BCFAs levels. Different dietary interventions, small study samples, and differences in the intestinal transit time related to IBS subtypes may explain inconsistencies and preclude the drawing of safe conclusions.

### 2.3. BCFAs as Potentially Harmful Metabolites

BCFAs have been proposed as markers of colonic protein fermentation, a process that leads to the production of nitrogenous metabolites of protein and amino acid fermentation, such as amines, hydrogen sulfide, p-cresol, phenols, and ammonia. These metabolites are toxic for colonocytes and they are associated with the development inflammatory conditions [29,57]. Windey et al. supported, with evidence, the hypothesis that these luminal protein end-products affect not only epithelial cell metabolism but also intestinal barrier function [58]. In a cross-sectional study, scientists compared the concentrations of BCFAs, ammonia, and fecal pH between vegans and omnivores. Results showed that protein intake was higher in omnivores than in vegans and there was a trend of lower BCFAs concentrations in vegans compared to omnivores, though no significant correlation with protein intake was detected [57]. Moreover, in an in vitro study [59], BCFA levels were lower in anaerobic incubation of vegetarians stools compared to incubation of fecal samples of omnivorous. Bacteria from vegetarian donors grew faster on soy protein as substrate, while in omnivorous donors, meat protein and casein were the preferred growth substrates. Fermentation patterns on different substrates were observed between the gut microbiota of vegetarians and omnivore donors. Differences were focused on BCFAs, ammonia, and total bacteria, with lower BCFA levels to be found in vegetarians; this suggests that these donors may have lower branched-chain amino acids intake [59]. The ability of protein end-products to have harmful effects on the GI is likely related to luminal concentrations of these metabolites. Metabolites’ concentrations may depend upon the balance between the rates of production, detoxification by the colonic epithelium, and absorption or excretion from the large intestine [25]. The protein source (animal- or plant-based) and the effect of food processing may play a role in altering its digestibility. The proportion of protein consumed and the presence of other nutrients may share a similar efficiency in digestibility and absorption [26]. Nevertheless, the evidence did not prove that protein fermentation is a risk factor for inflammation or bowel diseases (IBS, IBD, colorectal cancer, etc.) development. The impact of this fermentation may be affected by other dietary or lifestyle risk factors [58].

Even if changes in the diet constitute a factor affecting the BCFAs production, age can also be related to it. A physiological decline in the functionality of gut microbes has been highlighted over the years [60]. Additionally, higher levels of apoptotic cells in aging gut mucosa may allow more fermentable amino acids to be available to the microbiota, resulting in a higher level of BCFAs [29].

## 3. Conclusions

IBS is a difficult-to-manage disorder, and it is associated with poor health quality of life [1]. Diet and dietary end-products have been related to the development or the management of the syndrome. While most of the literature focuses on SCFAs, less is known about the way that BCFAs influence the development or dietary management of IBS. LFD is an established dietary intervention for amelioration of IBS symptoms; it has been associated with changes in of the gut microbiome, and due to its composition, it leads to higher protein and amino acid fermentation. Both animal studies and human clinical trials show that changes in the diet of fermentable, indigestible carbohydrates, like in LFD intervention, may lead to production of BCFAs, the byproduct of this fermentation, which, under certain circumstances, might produce large amounts of SCFAs that eventually may modulate and decrease intestinal inflammation [38].

Although evidence suggests that BCFAs might play a protective role in gut inflammation, other nitrogenous metabolites of protein fermentation, such as amines, hydrogen sulfide, p-cresol, phenols, and ammonia, have detrimental effects on colonocytes and they are associated with gut inflammation, a condition that has been pathogenetically associated with IBS [1]. Dietary modifications in protein intake by changing, for example, red meat consumption to white meat such as chicken and fish, or plant-based proteins, may reduce the availability of nitrites in the colon [25].

Further research on the topic must consider the lack of clinical studies in a sufficient sample of IBS-D patients, especially those receiving LFD, with simultaneous measurements of inflammatory biomarkers and fecal BCFAs level. Until new valid evidence accumulates, the role of BCFAs in IBS pathogenesis or symptoms management with LFD remains obscure.

## Figures and Tables

**Figure 1 microorganisms-11-02387-f001:**
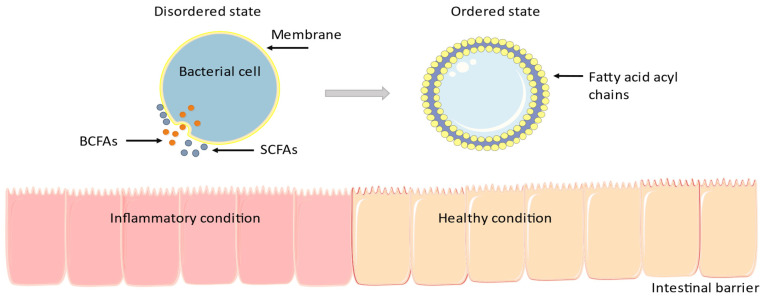
Many microorganisms use BCFAs in their membranes to modulate biophysical procedures [34]. BCFAs are taken up and incorporated into enterocyte membranes where they modulate the inflammatory response [35,36,37]. The figure was designed using Servier Medical Art, provided by Servier, licensed under a Creative Commons Attribution 3.0 unported license.

**Figure 2 microorganisms-11-02387-f002:**
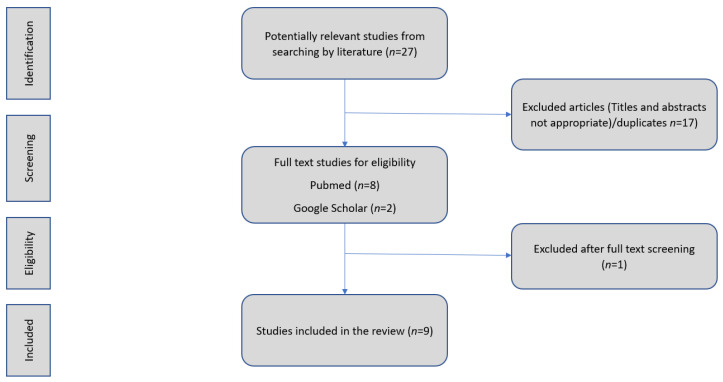
Flow chart. Identification and selection of the studies.

## Data Availability

Not applicable.

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
