# Peer review of "The Low-FODMAP Diet, IBS, and BCFAs: Exploring the Positive, Negative, and Less Desirable Aspects—A Literature Review"

_microorganisms, 2023, doi:10.3390/microorganisms11102387_

Round 1

Reviewer 1 Report

The manuscript entitled “Low FODMAP diet, IBS and BCFAs; the good the bad and the ugly. A literature review” aimed to discuss the relationship among Low FODMAP diet, branched short-chain fatty acids (BCFA) and irritable bowel syndrome (IBS). The researchers summarized the theoretical pathway of action of BCFA on IBS as well as Low FODMAP diet pathway regulation of IBS via BCFA and practical experimental investigations.

The English expressions throughout the review need to be strengthened and there are many grammatical errors. Some revisions of formatting and grammatical errors are list below.

Line 20 Replace “were” with “are”.

Line 47 Replace “BCFA’s” with “BCFAs”.

Line 48 Replace “present” with “presenting”.

Line 76-77 Replace “thus increase the amount of fluids in the intestines contributing to their osmotic activity” with “thus increasing the amount of fluids in the intestines and contributing to their osmotic activity”.

Line 89 Replace “and decrease” with “and a decrease”.

Line 94 Replace “secrete” with “secreting”.

Line 262 Replace “suggest” with “suggested”.

There are still some issues need to be improved.

Line92-96. The inflammation status and gut microbiota should be introduced. Please refer this reference (Food & Function, 2022, 13(24), 12686-12696. Food Chemistry, 417(2023):135861.).

Line 108-110

This sentence placed here is not relevant to the whole paragraph and is not relevant to the topic. Line 121

The following sentences are pretty wordy.

Line 129

“As a result” That's an inappropriate use of the word.

Line 173

Here pro-inflammatory and anti-inflammatory pathways should be placed above and integrated together.

Line 245

It is recommended that “A plausible speculation” not appear.

Line 296

Table 2 is a little less informative.

Line 167,253,353

No references in this section.

Line 381

The article is small and still a bit disorganized, it would have been better to add this part of the study.

The manuscript entitled “Low FODMAP diet, IBS and BCFAs; the good the bad and the ugly. A literature review” aimed to discuss the relationship among Low FODMAP diet, branched short-chain fatty acids (BCFA) and irritable bowel syndrome (IBS). The researchers summarized the theoretical pathway of action of BCFA on IBS as well as Low FODMAP diet pathway regulation of IBS via BCFA and practical experimental investigations.

The English expressions throughout the review need to be strengthened and there are many grammatical errors. Some revisions of formatting and grammatical errors are list below.

Line 20 Replace “were” with “are”.

Line 47 Replace “BCFA’s” with “BCFAs”.

Line 48 Replace “present” with “presenting”.

Line 76-77 Replace “thus increase the amount of fluids in the intestines contributing to their osmotic activity” with “thus increasing the amount of fluids in the intestines and contributing to their osmotic activity”.

Line 89 Replace “and decrease” with “and a decrease”.

Line 94 Replace “secrete” with “secreting”.

Line 262 Replace “suggest” with “suggested”.

There are still some issues need to be improved.

Line92-96. The inflammation status and gut microbiota should be introduced. Please refer this reference (Food & Function, 2022, 13(24), 12686-12696. Food Chemistry, 417(2023):135861.).

Line 108-110

This sentence placed here is not relevant to the whole paragraph and is not relevant to the topic. Line 121

The following sentences are pretty wordy.

Line 129

“As a result” That's an inappropriate use of the word.

Line 173

Here pro-inflammatory and anti-inflammatory pathways should be placed above and integrated together.

Line 245

It is recommended that “A plausible speculation” not appear.

Line 296

Table 2 is a little less informative.

Line 167,253,353

No references in this section.

Line 381

The article is small and still a bit disorganized, it would have been better to add this part of the study.

Author Response

Reviewer 1

  1. The English expressions throughout the review need to be strengthened and there are many grammatical errors. Some revisions of formatting and grammatical errors are list below.
  • Line 20 Replace “were” with “are”.
  • Line 47 Replace “BCFA’s” with “BCFAs”.
  • Line 48 Replace “present” with “presenting”.
  • Line 76-77 Replace “thus increase the amount of fluids in the intestines contributing to their osmotic activity” with “thus increasing the amount of fluids in the intestines and contributing to their osmotic activity”.
  • Line 89 Replace “and decrease” with “and a decrease”.
  • Line 94 Replace “secrete” with “secreting”.
  • Line 262 Replace “suggest” with “suggested”.

Response: We apologize for the errors. Corrections have been undertaken, as requested and the whole document has been checked for language, grammatical and syntax errors by a native English speaking dietician student listed in our acknowledgement.

Line92-96. The inflammation status and gut microbiota should be introduced. Please refer this reference (Food & Function, 2022, 13(24), 12686-12696. Food Chemistry, 417(2023):135861.).  

Response: The relation of Inflammation status with microbiota is now introduced in the sixth paragraph of the introduction, as requested.

Line 108-110.This sentence placed here is not relevant to the whole paragraph and is not relevant to the topic.  

Response: We removed the sentence, as requested.

Line 121 .The following sentences are pretty wordy.

Response: We hope that the text now is better readable following the grammatical polishing

Line 129 “As a result” That's an inappropriate use of the word.

Response: The expression was corrected by a native English-speaking person, as requested. 

Line 173 Here pro-inflammatory and anti-inflammatory pathways should be placed above and integrated together.

Response: Thank you for your comment. In this paragraph, we have collected all the studies that mention the role of BCFAs in intestinal inflammation. The results of each study are presented in a separate sub-paragraph. We suggest that this is more convenient for the reader to understand the different pathways (pro- or anti-inflammatory) depending on the details of each study.

Line 245 It is recommended that “A plausible speculation” not appear.

Response: The expression was corrected by a native English-speaking person, as requested. 

Line 296 Table 2 is a little less informative.  

Response: We are sorry to disagree. Table 2 summarizes the preclinical studies that have investigated the interactions between low FODMAP diet and BCFAs. Furthermore, the information presented in Table 2 shares the same philosophy with Tables 1 & 3.

Line 167,253,353 No references in this section.

Response: We included the requested references in these lines, although they also appear at the end of the relative paragraph, in the original submission.

Line 381 The article is small and still a bit disorganized, it would have been better to add this part of the study.

Response: We cannot understand the meaning of the comment referring to line 381. Sorry about this.

Reviewer 2 Report

The Authors reviewed systematically the papers to indicate and clarify the role of BCFAs on gut inflammation in patients with IBS and the impact of LFD in BCFAs production.

Although the study has the potentiality of being shared with the scientific community, I believe that the manuscript would benefit from a revision with the attempt to better support their experimental setting.

Develop this important parts:

Abstract: include purpose, methods, results, and conclusions of the report

Flow diagram: report the number of records identified from each database

Search strategy: specify the methods used to decide whether a study met the inclusion criteria of the review, including how many reviewers screened each record and each report retrieved, whether they worked independently.

Include PICOS and Quality Assesment

Specify the methods used to assess risk of bias in the included studies (with graph)

Provide registration information for the review, including register name and registration number, or state that the review was not registered

Result: add a separate result paragraph where describe the results of the search

Discussion: add a separate discussion paragraph where provide a general interpretation of the results in the context of other evidence

Author Response

Reviewer 2

Develop this important parts: Abstract: include purpose, methods, results, and conclusions of the report

Response: We fulfilled your request.

Flow diagram: report the number of records identified from each database

Response: Flow diagram has been updated, as requested.

Search strategy: specify the methods used to decide whether a study met the inclusion criteria of the review, including how many reviewers screened each record and each report retrieved, whether they worked independently.

Response: We fulfilled your request by updating the literature search paragraph.

Include PICOS and Quality Assesment

Specify the methods used to assess risk of bias in the included studies (with graph)

Provide registration information for the review, including register name and registration number, or state that the review was not registered  

Response: Our review is not a systematic review to fulfil these requirements (PICOS, quality assessment, registration number etc).

Result: add a separate result paragraph where describe the results of the search

Response: You can find the requested paragraph at page 12 of the revised manuscript (track changes form).

Discussion: add a separate discussion paragraph where provide a general interpretation of the results in the context of other evidence

Response: You may find the requested info at the revised 1st paragraph in the conclusions section.

Reviewer 3 Report

Nikolaki et al used this review paper to investigate the connection between branched short-chain fatty acids (BCFAs) and irritable bowel syndrome (IBS), shedding light on their roles and impacts. BCFAs, resulting from the bacterial breakdown of branched-chain amino acids, have been linked to colonic protein fermentation. IBS, characterized by gut inflammation and dysbiosis, often responds to the Low FODMAP diet (LFD), which reduces fermentable carbohydrates and short-chain fatty acid (SCFA) production. Meanwhile, elevated protein consumption increases BCFAs, which are associated with colonic microbiome changes and inflammation. The study explores how BCFAs affect gut inflammation in IBS and LFD's influence on BCFAs. The review aims to clarify the intricate relationship between BCFAs, inflammation, and dietary interventions for IBS management. This review paper is interesting to me. I am quite open to looking at a revised version if the authors could address some major and minor issues in a satisfactory fashion, which I describe in more detail below.

Major issues:

  1. Some sentences are quite confusing and I don’t quite understand their meanings. I believe making them more concise would improve readability. For instance, I don’t understand the sentence in lines 51-52 “BCFAs constitute a considerable component of a range as well of gut bacteria, about 15% of phyla”. Shall it be “BCFAs exist in the metabolic networks of many gut bacteria (~15% of phyla)”? Another example is in lines 55-56 “the essential amino acids’ valine, leucine, and isoleucine, respectively”. Maybe it should be “valine, leucine, and isoleucine, respectively (all belong to essential amino acids)”? More comments on grammatical errors can be found in the minor comments.

2.     The text appears to conclude by stating that the role of BCFAs in IBS pathogenesis or management remains unclear due to a lack of clinical studies. It might be helpful to reiterate the main points discussed and suggest potential avenues for further research.

3.     The mechanisms of how the diet modulates the gut microbiome and how the gut microbiome subsequently impacts metabolite production are not properly summarized. I would suggest an introduction to the relationships between foods, gut microbes, and metabolites. The authors may refer to the past works that have systematically captured their interactions that have been found in many pieces of literature (Akshit Goyal et al., Nature Communications 2021; Jaeyun Sung et al., Nature Communications 2017) and works of dietary intervention studies about BCFAs (Sharon Thompson, The Journal of Nutrition 2021; Hannah Holscher, The Journal of Nutrition 2018; Tong Wang et al., bioRxiv, 2023.03.14.532589).

Minor comments:

1.     Lines 2-3: I think the current title “Low FODMAP diet, IBS and BCFAs; the good the bad and the 2 ugly. A literature review” is grammatically wrong. I would suggest changing it to “The low FODMAP diet, IBS, and BCFAs: exploring the positive, negative, and less desirable aspects – a literature review”.

2.     Line 27: “detrimental effects to intestinal barrier” -> “detrimental effects on intestinal barrier”

  1. Line 51: “the last trimester human fetuses” -> “the last trimester of human fetuses”
  2. Line 63: “abundance of BCFAs” -> “the abundance of BCFAs”
  3. Line 76: “, thus increase” -> “, thus increasing”
  4. Line 99: “detrimental effects for gut health” -> “detrimental effects on gut health”
  5. Line 109: “causes muscle relaxation” -> “cause muscle relaxation”
  6. Line 126: “In details,” -> “In detail,”
  7. Line 203: “identified by manual search” -> “identified by the manual search”
  8. Line 222: “In another case control study,” -> “In another case-control study,”
  9. Line 374: “Although, evidence suggests” -> “Although evidence suggests”

Some sentences are quite confusing and I don’t quite understand their meanings. I believe making them more concise would improve readability. For instance, I don’t understand the sentence in lines 51-52 “BCFAs constitute a considerable component of a range as well of gut bacteria, about 15% of phyla”. Shall it be “BCFAs exist in the metabolic networks of many gut bacteria (~15% of phyla)”? Another example is in lines 55-56 “the essential amino acids’ valine, leucine, and isoleucine, respectively”. Maybe it should be “valine, leucine, and isoleucine, respectively (all belong to essential amino acids)”? More comments on grammatical errors can be found in the minor comments.

Author Response

Reviewer 3

Major issues:

1.Some sentences are quite confusing and I don’t quite understand their meanings. I believe making them more concise would improve readability. For instance, I don’t understand the sentence in lines 51-52 “BCFAs constitute a considerable component of a range as well of gut bacteria, about 15% of phyla”. Shall it be “BCFAs exist in the metabolic networks of many gut bacteria (~15% of phyla)”? Another example is in lines 55-56 “the essential amino acids’ valine, leucine, and isoleucine, respectively”. Maybe it should be “valine, leucine, and isoleucine, respectively (all belong to essential amino acids)”? More comments on grammatical errors can be found in the minor comments.  

Response: Requested corrections have been undertaken and the whole document has been checked for language, grammatical and syntax errors by a native English speaking student dietician listed in our acknowledgement.  

2.The text appears to conclude by stating that the role of BCFAs in IBS pathogenesis or management remains unclear due to a lack of clinical studies. It might be helpful to reiterate the main points discussed and suggest potential avenues for further research.

Response: In our revision, we summarize the main points discussed in the manuscript in the 1st and second paragraph of the conclusions section.

  1. The mechanisms of how the diet modulates the gut microbiome and how the gut microbiome subsequently impacts metabolite production are not properly summarized. I would suggest an introduction to the relationships between foods, gut microbes, and metabolites. The authors may refer to the past works that have systematically captured their interactions that have been found in many pieces of literature (Akshit Goyal et al., Nature Communications 2021; Jaeyun Sung et al., Nature Communications 2017) and works of dietary intervention studies about BCFAs (Sharon Thompson, The Journal of Nutrition 2021; Hannah Holscher, The Journal of Nutrition 2018; Tong Wang et al., bioRxiv, 2023.03.14.532589).

Response: Thank you for your remark. A summary of the diet-microbiome-metabolites interaction is now presented in the revised paragraph 4 of the introduction.

Minor comments:

  1. Lines 2-3: I think the current title “Low FODMAP diet, IBS and BCFAs; the good the bad and the 2 ugly. A literature review” is grammatically wrong. I would suggest changing it to “The low FODMAP diet, IBS, and BCFAs: exploring the positive, negative, and less desirable aspects – a literature review”.

Response: We accept your elegant suggestion. Thank you!

  1. Line 27: “detrimental effects to intestinal barrier” -> “detrimental effects on intestinal barrier”
  2. Line 51: “the last trimester human fetuses” -> “the last trimester of human fetuses”
  3. Line 63: “abundance of BCFAs” -> “the abundance of BCFAs”
  4. Line 76: “, thus increase” -> “, thus increasing”
  5. Line 99: “detrimental effects for gut health” -> “detrimental effects on gut health”
  6. Line 109: “causes muscle relaxation” -> “cause muscle relaxation”
  7. Line 126: “In details,” -> “In detail,”
  8. Line 203: “identified by manual search” -> “identified by the manual search”
  9. Line 222: “In another case control study,” -> “In another case-control study,”
  10. Line 374: “Although, evidence suggests” -> “Although evidence suggests”

 Response: Requested corrections have been undertaken and the whole document has been checked for language, grammatical and syntax errors by a native English speaking student dietician listed in our acknowledgements.

Round 2

Reviewer 1 Report

The reference should be updated with the reviewer's comment. It can be accepted in other issues. 

The reference should be updated with the reviewer's comment. It can be accepted in other issues. 

Reviewer 3 Report

The authors have addressed all my concerns. I don’t have other comments.